# Evolving Embodied Intelligence: Graph Neural Network–Driven Co-Design of Morphology and Control in Soft Robotics

## Abstract

The intelligent behavior of robots does not emerge solely from control systems, but from the tight coupling between body and brain—a principle known as embodied intelligence. Designing soft robots that leverage this interaction remains a significant challenge, particularly when morphology and control require simultaneous optimization. A significant obstacle in this co-design process is that morphological evolution can disrupt learned control strategies, making it difficult to reuse or adapt existing knowledge. We address this by develop a Graph Neural Network-based approach for the co-design of morphology and controller. Each robot is represented as a graph, with a graph attention network (GAT) encoding node features and a pooled representation passed through a multilayer perceptron (MLP) head to produce actuator commands or value estimates. During evolution, inheritance follows a topology-consistent mapping: shared GAT layers are reused, MLP hidden layers are transferred intact, matched actuator outputs are copied, and unmatched ones are randomly initialized and fine-tuned. This morphology-aware policy class lets the controller adapt when the body mutates. On the benchmark, our GAT-based approach achieves higher final fitness and stronger adaptability to morphological variations compared to traditional MLP-only co-design methods. These results indicate that graph-structured policies provide a more effective interface between evolving morphologies and control for embodied intelligence.

## 1 Introduction

Developing autonomous agents that operate reliably in complex, dynamic environments is the most important goal of artificial intelligence and artificial life. Soft robots, built from highly compliant materials such as polymers, elastomers, or silicone, provide distinct benefits for safe human interaction and adaptable locomotion in everyday environments Marchese (2015). From the perspective of embodied intelligence Pfeifer et al. (2007), their bodies are not passive geometry but integral parts of computation: morphology, materials, and control jointly shape behavior. This flexibility makes design difficult, leading to costly controller optimization to accurately predict soft-body dynamics and typically achieve reliable adaptation van Diepen & Shea (2022).

A growing line of work seeks to co-design morphology and control, echoing the biological co-evolution of body and brain Bhatia et al. (2021); Cheney et al. (2013); Corucci et al. (2018; 2016); Van Diepen & Shea (2019). However, two obstacles limit scalability to harder tasks: (i) substantial training cost as each new morphology typically starts control learning from scratch, and (ii) fragile controller inheritance across generations, since changes in sensor/actuator layouts break the fixed-input assumptions of conventional multi-layer perceptron (MLP) policies Bhatia et al. (2021). Evolutionary attempts at policy transfer Tanaka & Aranha (2022) and DRL-based Lamarckian inheritance Harada & Iba (2024) help, but remain constrained by architecture mismatch and ad-hoc transfer rules.

We overcome these limitations with a morphology-aware policy representation, shown in Figure 1. Robots are modeled as graphs, where nodes correspond to functional components (e.g., sensors, actuators, voxels) and edges encode structural or kinematic relations. Controllers are implemented as Graph Attention Networks (GATs) trained with DRL: node embeddings are aggregated and passed

through a lightweight MLP head that generates actuator commands or value estimates. When morphology mutates, the graph reconfigures naturally, and policies transfer through embedding reuse, shared MLP weights, and topology-consistent mapping rather than fixed input indices. Coupling this inheritance with a Genetic Algorithm (GA) and GAT-based PPO yields a structure-aware co-design framework, enabling offspring to inherit and adapt parental policies efficiently. This reduces retraining cost and improves robustness to morphological variations in EvoGym tasks.

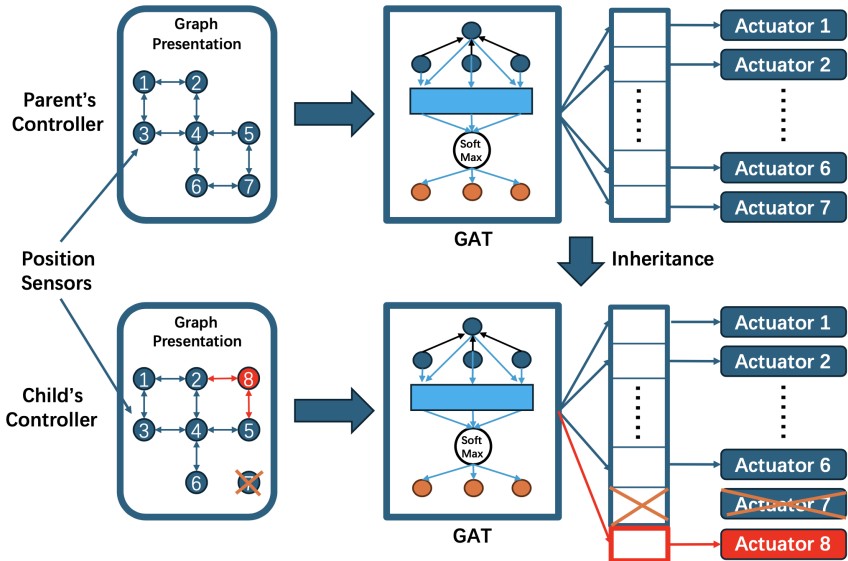

Figure 1: Overview of the proposed GAT-based policy framework with DRL inheritance. The parent controller (top) represents the robot as a graph, where nodes denote position sensors and edges capture spatial relationships. A GAT encodes node features, which are pooled into a fixed-length vector and passed through a lightweight MLP head to generate actuator control signals. During inheritance (bottom), the trained controller is transferred to the child. When morphology changes, connections to removed actuators are discarded, and new ones are initialized for added actuators.

Our main contributions are:

- A co-design algorithm that integrates GAT-based policies with DRL to realize *morphology-aware* controller inheritance. An embedding-level transfer scheme that accelerates adaptation of offspring controllers by reusing graph features aligned to new morphologies.

- A graph representation that preserves policy competence under structural mutation, overcoming fixed-input MLP limitations in co-design.

- Empirical validation on benchmark platform showing higher final rewards, and improved robustness to morphology changes versus MLP-based baselines, with ablations isolating the effects of graph policies and inheritance.

Together, these results support the claim that graph-structured policies provide an effective interface between evolving bodies and brains: they operationalize embodied intelligence in co-design by coupling morphology and control through shared structure, and they offer a scalable path to soft-robot agents that learn more efficiently while generalizing across diverse, changing morphologies.

## 2 BACKGROUND

### 2.1 GENETIC ALGORITHM AND PROXIMAL POLICY OPTIMIZATION

Genetic Algorithms (GAs) Mitchell (1998) are population-based optimization methods inspired by natural selection, where candidate solutions evolve through selection, mutation, and crossover. They are effective for exploring large, non-convex design spaces and have been widely applied in evolutionary robotics. Reinforcement learning (RL) Sutton & Barto (1998) offers a complementary

paradigm, training agents to maximize cumulative rewards through interaction with their environment. Among RL methods, Proximal Policy Optimization (PPO) Schulman et al. (2017) is especially popular in robotics for its simplicity, stability, and strong empirical performance. PPO follows an actor–critic scheme, alternating between trajectory collection with the current policy and updates using a clipped surrogate objective that avoids destabilizing policy shifts.

## 2.2 GRAPH NEURAL NETWORKS AND GRAPH ATTENTION NETWORKS

Graph Neural Networks (GNNs) Xu et al. (2019); Wu et al. (2021) generalize deep learning to graph-structured inputs by refining node features through iterative message passing with neighbors. This makes them well-suited for modular robots, which can be naturally modeled as graphs of connected components. In contrast to MLPs, GNNs can flexibly handle morphological variations without altering the underlying architecture. Graph Attention Networks (GATs) Velickovic et al. (2018) extend this framework by introducing attention over edges, enabling the model to learn which connections are most important for control.

## 2.3 BENCHMARK PLATFORM

Several studies have explored soft robot co-design in simulation Cheney et al. (2013); Corucci et al. (2018; 2016); Van Diepen & Shea (2019), but progress has been limited. Controllers are often reduced to simple, repetitive actuation, and tasks typically involve only basic locomotion, making adaptation to complex environments difficult. Comparisons across methods are also hindered by reliance on custom evaluation setups Bhatia et al. (2021). To overcome these challenges, Bhatia et al. (2021) introduced Evogym, a standardized 2D simulation platform. Robots are constructed from four voxel types, including: rigid (black), soft (gray), horizontal actuators (light blue), and vertical actuators (orange), and equipped with sensors for position, velocity, rotation, and object interaction (see Figure 2a). Position sensors (grey dots in Figure 2a), always included, scale with morphology since they are tied to voxel vertices, while the other sensors are single-instance. Actuators (green-outlines colored voxels in Figure 2b) receive continuous control signals that govern the contraction or extension of their components. Further details can be found in Bhatia et al. (2021); Harada & Iba (2024).

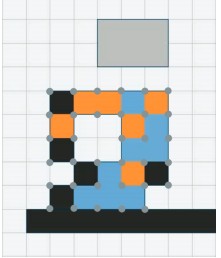 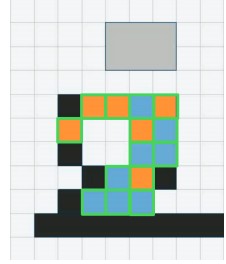 

(a) structure with position sensors     (b) structure with actuators     (c) GA-based robot's genome

Figure 2: Illustration of a soft robot designed for the Thrower-v0 task in the Evogym framework.

## 3 METHODOLOGY

**Co-Design with GAT Controllers** We present a co-design algorithm for soft robots in which morphology and control evolve together. Unlike approaches that retrain controllers from scratch, our method enables direct inheritance of policies learned through DRL, allowing offspring to build on the experience of their parents. This inheritance accelerates adaptation and promotes the emergence of robots capable of solving more complex tasks. The overall procedure is outlined in Algorithm 1, with mutation and selection following the evolutionary framework of Bhatia et al. (2021).

**Graph-Based Controller Representation** Each robot is represented as a graph $G = (V, E)$, where nodes correspond to position sensors and edges capture spatial adjacency. Nodes are assigned feature vectors that combine global properties (e.g., orientation) with local information (e.g., coordinates, voxel type, and velocity).

---

**Algorithm 1** Co-Design with GAT-based Controllers

---

**Require:** population size $p$, max generations $n$
1: **Init:** For $k = 1 \dots p$, sample morphology $\mathcal{M}_k$, build graph $G_k = (V_k, E_k)$, init actor $\pi_k(\cdot \mid x, G_k; \theta_k^{\text{act}})$ and critic $V_k(x, G_k; \theta_k^{\text{crt}})$
2: **for** $g = 1 \dots n$ **do**
3:     **for** $k = 1 \dots p$ **do**
4:         **if** $\mathcal{M}_k$ **is newborn then**
5:             Train $(\theta_k^{\text{act}}, \theta_k^{\text{crt}})$ with DRL (e.g., PPO) on environment $\mathcal{E}_k$
6:             $f_k \leftarrow$ best episodic return of $(G_k, \pi_k)$
7:         **else**
8:             Retain fitness $f_k$ from previous generation
9:         **end if**
10:     **end for**
11:     **Selection:** keep top-$m$ elites by $f_k$; mark non-elites as dead
12:     **for** each dead slot $k$ **do**
13:         Choose parent $u$ from elites randomly
14:         $\mathcal{M}_k \leftarrow \text{MUTATEMORPH}(\mathcal{M}_u)$
15:         $G_k \leftarrow \text{BUILDGRAPH}(\mathcal{M}_k)$
16:         $(\theta_k^{\text{act}}, \theta_k^{\text{crt}}) \leftarrow \text{MAPWEIGHTS}(\theta_u^{\text{act}}, \theta_u^{\text{crt}}, G_u, G_k)$         ▷ Co-design inheritance
17:         mark $\mathcal{M}_k$ as newborn
18:     **end for**
19: **end for**
20: $\ell \leftarrow \arg\max_k f_k$
21: **return** $(\mathcal{M}_\ell, G_\ell, \pi_\ell(\cdot \mid x, G_\ell; \theta_\ell^{\text{act}}))$

---

Unlike MLPs, which flatten inputs into a fixed vector and rely on a centralized controller, GNNs model robots as interconnected components, allowing actuators to act locally while obtaining global sensor and actuator information from their neighboring nodes through message passing. This decentralized structure scales naturally, as morphological changes such as adding or removing actuators can be incorporated without redesigning the policy. Within this family, GATs offer an additional advantage by learning attention weights that highlight the most relevant connections, improving generalization across morphologies and adaptation to structural changes. Attention also helps the policy identify how specific sensor–actuator interactions shape movement, enabling GAT-based controllers to combine robustness to variation with the flexibility to support diverse locomotion and manipulation strategies.

---

**Algorithm 2** MAPWEIGHTS for GAT-based Actor/Critic with Morphology Changes

---

**Require:** Parent weights $(\theta_u^{\text{act}}, \theta_u^{\text{crt}})$, parent graph $G_u = (V_u, E_u)$, child graph $G_k = (V_k, E_k)$
1: Compute node correspondence $\mathcal{C} : V_k \to V_u \cup \{\varnothing\}$ by spatial matching
2: **Actor:**
3:     Copy all shared GAT message-passing layers (attention and linear kernels) from $\theta_u^{\text{act}}$ to $\theta_k^{\text{act}}$ ▷ hidden layers fully inherited
4:     Copy hidden layers of the pooled MLP head in full
5:     **For each actuator in the final output layer:**
6:         **if** actuator matches a parent actuator **then** copy corresponding weights
7:         **else if** new actuator **then** initialize weights randomly
8:         **else if** actuator removed **then** discard weights
9: **Critic:**
10:     Copy shared GAT layers from $\theta_u^{\text{crt}}$ to $\theta_k^{\text{crt}}$
11:     Copy global pooling and all hidden MLP layers in full
12:     Keep final scalar output head identical (critic output dimension is invariant)
13: **Return** $(\theta_k^{\text{act}}, \theta_k^{\text{crt}})$

---

We investigate two strategies for constructing node features: **(i) GA-GAT-PPO-Global-Transfer**, where node features are averaged and assigned uniformly to all nodes, and **(ii) GA-GAT-PPO-Local-Transfer**, where each node is given its own feature vector. To capture spatial structure, edge

features are extended with relative offsets $(\Delta x, \Delta y)$, enabling the controller to attend to both node attributes and their geometric relations. The resulting graph is processed by a GAT layer, which aggregates information through one attention-based message passing round, followed by averaging over nodes. The average representation is then fed into a lightweight MLP head: its hidden layers are shared across morphologies, while the output layer maps to actuator commands for the actor or a scalar value estimate for the critic.

**Inheritance of GAT Controllers** Robot morphologies evolve through mutations that can add, remove, or alter voxels, thereby changing the set of sensors and actuators. To transfer controllers across such changes, we introduce the MAPWEIGHTS procedure (Algorithm 2), which maps parameters from parent to child according to the following rules:

- Shared hidden layers: The GAT's message-passing and attention layers are inherited in full across morphologies, and the hidden layers of the output MLP are also fully reused.

- Actuator mapping: For the final actuator layer, weights connected to matched actuators are inherited from the parent, new actuators are initialized randomly, and removed actuators are discarded.

The critic network inherits parameters in the same way. Since its output is always a single scalar, the final prediction layer remains unchanged across morphologies, while global pooling and the shared MLP hidden layers maintain compatibility with varying node counts. Taken together, Algorithms 1 and 2 define the GAT-based co-design cycle: robots are trained or evaluated, elites are selected, new morphologies are generated through mutation, and their controllers are initialized by mapping weights from parents to offspring. This process preserves learned representations across generations while adapting to structural changes, thereby speeding up co-design and enhancing performance.

## 4 EXPERIMENTAL SETUP

We conducted experiments in the EvoGym environment Bhatia et al. (2021) across four representative tasks, evaluating the following configurations: (i) our proposed method that combines a GA with a GAT-based PPO controller using inheritance under evolution, where global mean representations are shared across all nodes; (ii) our proposed method under the same setting, but with GAT-based PPO controllers that employ individualized node features incorporating each node's position and local state; (iii) a prior approach that applies inheritance to MLP-based PPO controllers under evolution Harada & Iba (2024); and (iv) a baseline genetic algorithm with PPO, where each new individual's controller is trained from scratch Bhatia et al. (2021). In all four settings, offspring are produced exclusively through mutation-only evolution.

The four tasks are briefly summarized below, with full descriptions available in Bhatia et al. (2021).

- **Pusher-v1**: The robot is required to push or drag a box placed behind it in the forward direction. This is a medium-difficulty task, and 700 robots are trained.

- **Thrower-v0**: The robot must throw a box that is initially positioned on top of it. This is also of medium difficulty, with 500 robots trained.

- **Carrier-v1**: The robot's objective is to transport a box to a table and place it on top. This is a hard task, with 500 robots trained.

- **Catcher-v0**: The robot attempts to catch a fast-moving, rotating box. This is considered a hard task, and 500 robots are trained.

To ensure a fair comparison, the hyperparameters for GA and PPO, as well as the survival rate and the probability of mutation, are adopted from Harada & Iba (2024). The number of robots trained per task, which also defines the number of generations, follows the experimental protocol outlined in Bhatia et al. (2021). For PPO, we rely on the publicly available implementation provided in Kostrikov (2018).

# 5 RESULT AND DISCUSSION

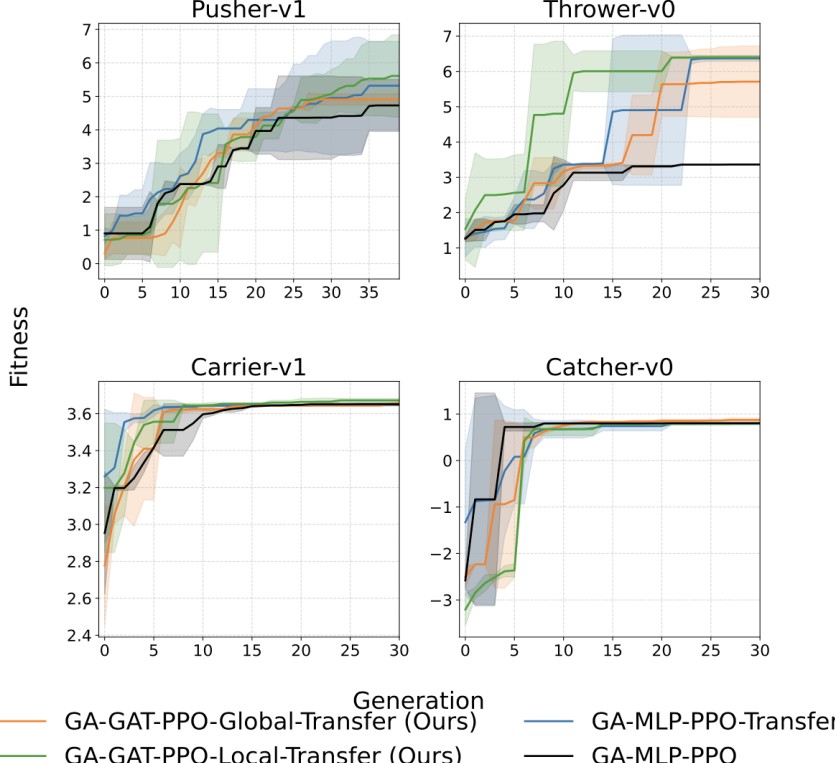

Figure 3: Impact of inheritance on evolution. We examine the influence of inheritance on evolutionary progress by tracking the fitness of the top-performing robot across generations. Each curve shows the mean performance over three independent runs, with shaded regions representing the standard deviation. Our GAT-based inheritance methods achieve higher peak fitness than baselines, with reduced variance across runs. GA-GAT-PPO-Local-Transfer, which provides individualized node representations, outperforms on Pusher-v1, Thrower-v0, and Carrier-v1, where localized coordination is critical. In contrast, GA-GAT-PPO-Global-Transfer, which employs a shared mean representation, performs best on Catcher-v0, a task that requires broader system-level coordination.

## 5.1 OVERALL RESULT ANALYSIS

Figure 3 presents the best fitness achieved across four representative tasks, illustrating how inheritance shapes evolutionary progress. Each curve reports the highest fitness per generation, averaged over three trials, with shaded bands indicating standard deviation. Our GAT-based approaches (GA-GAT-PPO-Global-Transfer and GA-GAT-PPO-Local-Transfer) consistently match or surpass the performance of MLP-based baselines. By exploiting attention to capture structural dependencies, they enable more effective policy transfer under morphological changes, resulting in stronger generalization and lower variance across runs.

In Pusher-v1 and Thrower-v0, both GAT-based variants reach higher peak fitness than GA-MLP-PPO, with the local feature design even outperforming GA-MLP-PPO-Transfer. In Thrower-v0, convergence is also faster in the early generations, showing that attention-guided inheritance accelerates learning by transferring useful traits more effectively. By comparison, MLP-only methods display higher variance, underscoring the stability advantage of our approach. In Carrier-v1 and Catcher-v0, the gains are most visible in robustness: both GAT variants rapidly attain near-optimal performance with consistently low variance, demonstrating that attention mechanisms improve not only performance but also reliability in evolutionary outcomes.

A task-level analysis further illustrates the complementary strengths of local versus global attention. Tasks requiring fine-grained, component-level coordination—such as Pusher-v1 (pushing or dragging a box), Thrower-v0 (sequentially launching a box), and Carrier-v1 (manipulating and transporting an object)—favor GA-GAT-PPO-Local-Transfer, which provides individualized node representations. Conversely, Catcher-v0, which demands rapid, system-wide synchronization to intercept a rotating object, benefits more from GA-GAT-PPO-Global-Transfer, where a shared global representation aligns behaviors across the body. These results suggest that local attention excels in tasks dominated by detailed part-level interactions, while global attention is more effective for behaviors requiring whole-body coordination.

Taken together, these results highlight the strengths of attention-based inheritance, including greater robustness, adaptability across different task requirements, and lower variability during training. More broadly, they suggest that inheritance guided by attention provides a scalable and principled foundation for evolutionary robotics, with potential extensions to more complex morphologies, multi-agent settings, and hybrid strategies that integrate both local and global reasoning. We attribute the superior performance of our approach compared with MLP baselines to two key factors: first, inheritance reduces the training burden by accelerating the acquisition of complex behaviors; second, it preserves morphological flexibility, allowing body structures to evolve freely without being restricted by the control policy.

## 5.2 CONTROLLER EVOLUTION ANALYSIS

Figure 4 compares the performance of four approaches on the Thrower-v0 task in EvoGym, where the main goal of the task is that the robot must catch a falling box and throw it as far as possible. Our proposed methods, GA-GAT-PPO-Global-Transfer (fitness score: 6.079) and GA-GAT-PPO-Local-Transfer (fitness score: 6.258), achieve substantially higher performance than the baselines. Both GAT-based variants develop stable and coordinated motion strategies that allow for consistent and effective throws toward the target. Among them, the local transfer method produces the most accurate throws, reliably reaching the target, while the global transfer method also succeeds but sometimes causes the box to rebound slightly after landing.

Under the same seed, the baseline methods GA-MLP-PPO-Transfer (fitness score: 3.268) and GA-MLP-PPO (fitness score: 3.353) struggle to produce consistent throwing strategies. Their best-performing robots often execute a high jump but quickly lose momentum, causing the box to fall short. By contrast, our GAT-based co-designed robots display motion patterns that resemble human-like throwing mechanics, making use of two actuators instead of the single actuator typically used in the baseline designs. This results in stronger propulsion and more effective throws, underscoring the benefits of attention-driven inheritance for complex manipulation tasks.

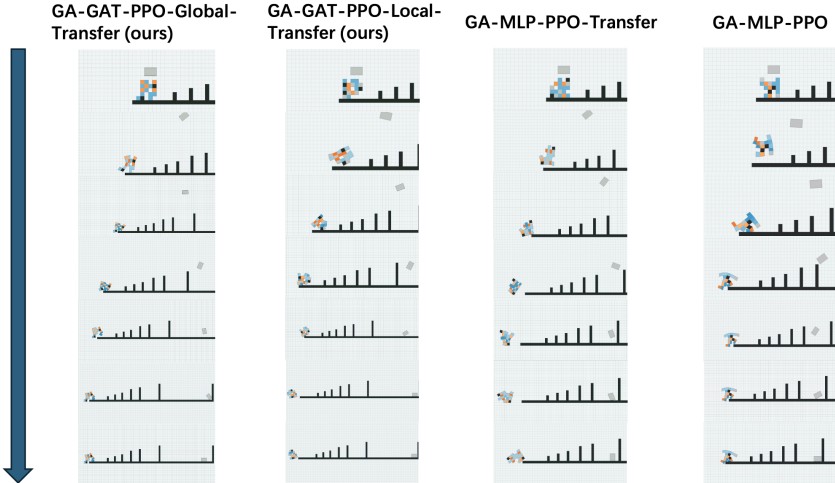

Figure 4: Visual comparison of four approaches applied to the Thrower-v0 task.

### 5.3 MORPHOLOGY EVOLUTION ANALYSIS

A key limitation of MLP-based PPO in co-evolving morphology and control is the fragility of inheritance under major structural changes. Since MLPs require fixed input and output dimensions, adding or removing voxels alters the parameter space and control mapping, often making inherited weights ineffective Harada & Iba (2024). GAT-based controllers overcome this by modeling the body as a graph, using message passing that adapts to new node configurations and attention to emphasize the most relevant connections. The pooled features are then processed by a lightweight MLP head to produce actuator commands or value estimates. This design combines parameter sharing, local reasoning, and adaptive attention, enabling robust performance even under significant morphological variation.

Figure 5 shows that, across all methods, the evolved robots tend to converge toward broadly similar morphologies, regardless of whether controllers are based on MLPs or GATs or whether inheritance is applied. While the exact voxel layouts vary, the designs consistently develop task-specific functional patterns, such as grasp-like forms in Carrier and extended appendages in Thrower. This outcome indicates that task requirements strongly shape the space of feasible morphologies, whereas the controller architecture mainly influences learning speed and adaptability rather than the overall class of final designs.

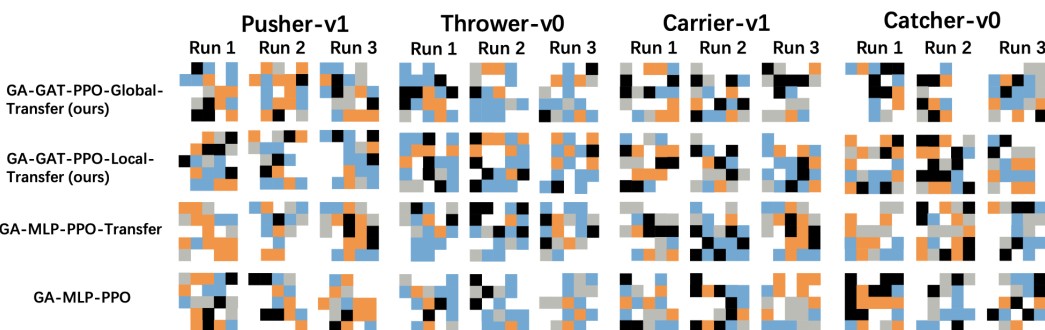

Figure 5: Comparison of evolved morphologies. For each method and trial, we illustrate the robot structures that achieved the highest fitness.

## 6 RELATED WORKS

### 6.1 CO-DESIGN OF MORPHOLOGY AND CONTROL

Joint optimization of body and controller dates back to Sims' virtual creatures, where articulated morphologies and MLP controllers co-evolved under task-driven fitness Sims (1994). Subsequent studies extended co-design to rigid robots Ha (2019); Pathak et al. (2019); Wang et al. (2019); Zhao et al. (2020), but limited degrees of freedom motivate soft-robot co-design with compliant materials Lee et al. (2017); Rus & Tolley (2015). In standardized evaluations on Evolution Gym (Evo-Gym) Bhatia et al. (2021), structure optimizers (GA, CPPN-NEAT, Bayesian) paired with PPO indicate GA as a strong morphology searcher, while PPO outperforms CPPN-based controllers across most tasks Saito et al. (2022). CPPN/HyperNEAT encodings generate spatially regular morphologies (and sometimes controllers) Stanley (2007); Stanley & Miikkulainen (2002); Tanaka & Aranha (2022); Cheney et al. (2013), but purely evolutionary controller optimization typically lags gradient-based RL. A recurring challenge across co-design pipelines is *controller reuse*: as morphology mutates, sensor/actuator layouts change, making fixed-shape MLP policies brittle and forcing expensive retraining.

### 6.2 MORPHOLOGY-AWARE TRANSFER AND GRAPH-STRUCTURED POLICIES

Transfer RL formalizes reuse of knowledge across tasks or embodiments, targeting jump-start and asymptotic gains even when state–action spaces differ Sutton & Barto (1998); Taylor & Stone (2005); Lazaric (2012). In soft-robot co-design, Lamarckian approaches transfer DRL controllers

across generations in EvoGym Harada & Iba (2024), but fixed-architecture MLPs remain brittle under I/O topology shifts. Graph-structured policies provide a part–relation inductive bias and can generalize over structural variation: NerveNet learns policies on graphs of body parts Wang et al. (2018), and graph-network models support inference/control with object–relation structure Sanchez-Gonzalez et al. (2018). Kurin et al. ("My Body Is a Cage") report, in incompatible MuJoCo control, that explicit morphological graphs do not always help over fully connected attention and introduce a Transformer controller that outperforms Graph Neural Network (GNN) baselines by sidestepping multi-hop message passing Kurin et al. (2021). Our setting differs in two key respects: (i) voxelized *soft* robots in EvoGym where morphology changes alter both sensors and actuators, and (ii) a *Lamarckian, topology-consistent* inheritance mechanism that maps actuator heads via graph correspondences. We show that attention-based GNN controllers with per-actuator heads, combined with structure-aware weight mapping and brief PPO adaptation, yield robust inheritance under morphological mutation on standardized EvoGym tasks.

## 7 CONCLUSION

In this paper, we propose a co-design algorithm that integrates GAT-based policies with DRL to enable morphology-aware controller inheritance. In standard PPO frameworks, the actor is typically modeled with an MLP, which assumes a fixed morphology and must be reinitialized or adapted with ad-hoc rules when structures change. Our approach overcomes this limitation by modeling each robot as a graph, allowing the controller to naturally handle variations in sensor count and connectivity. Shared attention layers and global pooling promote generalization across morphologies, while inheritance ensures that offspring retain and adapt parental knowledge after structural modifications. This design leads to stronger performance than MLP-only baselines.

Although GAT controllers often achieve higher final performance than MLP baselines, they do not always converge as quickly. Their greater architectural complexity requires learning both control policies and relational information through attention and message passing, which can slow early optimization. In addition, inheritance under morphological changes may introduce mismatches, as newly added nodes or edges are initialized without prior knowledge, causing temporary instability. By contrast, MLP controllers operate on fixed-length inputs and readily capture simple correlations, leading to faster early gains but weaker long-term generalization.

Looking ahead, several strategies could improve the efficiency and adaptability of GAT-based controllers. Training stability might be enhanced through attention regularization or curricula that introduce morphological changes gradually. Promising extensions include lightweight GAT variants that lower computational cost, as well as unified graph representations that model both sensors and actuators to capture perception–action coordination. Hybrid approaches that combine a lightweight MLP for rapid adaptation with a GAT for structural reasoning, or transfer methods such as knowledge distillation from simpler controllers, may further reconcile fast convergence with long-term generalization. Collectively, these directions could unite the quick learning of MLP-based controllers with the robustness and flexibility of GATs, enabling more efficient and scalable co-design algorithms.

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

## REPRODUCIBILITY STATEMENT

We have made several efforts to ensure the reproducibility of our work. The complete implementation of our proposed algorithm is available in an anonymous repository at `https://anonymous.4open.science/r/GNN-transfer-Soft-Robots-2026-ICLR-1233`. Section 4 provides detailed descriptions of the EvoGym tasks, implementation details, and hyperparameter settings, while Section 3 presents the algorithmic procedures in pseudocode. These resources collectively allow all reported results to be independently verified and extended.

## USE OF GENERATIVE AI

We used a GPT-based assistant (ChatGPT) exclusively for language editing (e.g., grammar, clarity, and concision) on draft text. The assistant did not generate research ideas, methods, analyses, results, figures, or data. All scientific content and conclusions are the authors' own. AI-suggested edits were reviewed and revised by the authors, who accept full responsibility for the manuscript. No confidential or sensitive data were provided to the tool.

