# OpenReview forum: "Evolving Embodied Intelligence: Graph Neural Network–Driven Co-Design of Morphology and Control in Soft Robotics"
_ICLR.cc/2026/Conference — Submitted to ICLR 2026_

### Official Review · Reviewer_du33 · 2025-10-17

**Soundness:** 1
**Presentation:** 1
**Contribution:** 1
**Rating:** 0
**Confidence:** 5

**Summary:**

The authors propose a graph attention controller — and a means of controller inheritance — in the context of robot morphology evolution. The authors claim that this method remedies disruptions in the reuse and adaptation of controllers as morphologies change during evolution. They evaluate their method and claims in the EvoGym environment.

**Strengths:**

The problem of simultaneously optimizing morphology and control is a longstanding unsolved problem. This is an important scientific and engineering problem and highly relevant to ICLR.

**Weaknesses:**

No form of hypothesis testing was reported (only mean fitness + stdev curves), but as far as I can tell, in all but one experiment, there was no difference with and without weight inheritance. In the one case where weight inheritance might affect fitness, there was no difference with and without the proposed method.

More importantly, there is no evidence that design optimization occurred. And there is no evidence that weights were inherited across different designs.

Finally, the scholarship is poor. I expected to see (at the very least some subset of) the following literature referenced/discussed:

1. Huang et al. One policy to control them all: Shared modular policies for agent-agnostic control. ICML (2020).
2. Yuan et al. Transform2Act: Learning a transform-and-control policy for efficient agent design. ICLR (2022).
3. Gupta et al. Metamorph: Learning universal controllers with transformers. ICLR (2022).
4. Hong et al. Structure-aware transformer policy for inhomogeneous multi-task reinforcement learning. ICLR (2022).
5. Xiong et al. Universal morphology control via contextual modulation. ICML (2023).
6. Xiong et al. Distilling morphology-conditioned hypernetworks for efficient universal morphology control. ICML (2024).
7. Strgar et al. Evolution and learning in differentiable robots. RSS (2024).
8. Mertan & Cheney. Towards multi-morphology controllers with diversity and knowledge distillation. GECCO (2024).
9. Li et al. Generating freeform endoskeletal robots. ICLR (2025).
10. Lu et al. Bodygen: Advancing towards efficient embodiment co-design. ICLR (2025).

**Questions:**

1. Based on evidence provided it is not clear the proposed method does what the authors claim. Why do the authors conclude their method improves fitness?

---

### Official Review · Reviewer_71DV · 2025-10-25

**Soundness:** 2
**Presentation:** 2
**Contribution:** 1
**Rating:** 2
**Confidence:** 4

**Summary:**

This paper introduces GNN for soft-robotics co-design and also introduce a scheme for morphology matching. Also to my best knowledge, GNN and Transformer are already heavily used for robot co-design.

**Strengths:**

This paper introduces a topology-consistent, graph-based policy representation that enables seamless controller inheritance across evolving morphologies, significantly improving adaptability and efficiency in soft robot co-design.

**Weaknesses:**

I have a relatively big concern about the contribution of this paper. GNN and Transformer are already heavily used in co-design, for example [1,2,3,4], what is the main difference between this work with them?

For example, paper [4] discusses the morphology representation for Transformer in co-design, and introduces a light-weight embedding for efficient morphology representation and morphology aligning using one network. GNN demonstrates little advantage over Transformer with a reasonable morphology embedding.

[1] Preco: Enhancing generalization in co-design of modular soft robots via brain-body pre-training

[2] Curriculum-based Co-design of Morphology and Control of Voxel-based Soft Robots

[3] Transform2act: Learning a transform-and-control policy for efficient agent design

[4] Bodygen: Advancing towards efficient embodiment co-design

**But I find the authors do not compare this work with them even though the topic is closely related. If possible, could the author provide any comparison with this line of works?**

**Questions:**

See weakness

---

### Official Review · Reviewer_WRgZ · 2025-10-28

**Soundness:** 3
**Presentation:** 2
**Contribution:** 2
**Rating:** 4
**Confidence:** 5

**Summary:**

This paper models soft/voxel robots as graphs (nodes represent voxels/actuators, edges represent adjacency/structural constraints) and constructs an actor-critic controller using graph attention networks (GATs). The outer layer employs genetic algorithms (GA) to explore morphologies, while the inner layer optimizes control via PPO. To address the long-standing issue of "morphological changes → control mismatch," the authors propose topology-consistent parameter inheritance/mapping: shared encoding layer weights are directly reused across morphologies, output channels are aligned by parent-child actuator pairs, and newly added/deleted components undergo minimal reset and fine-tuning. This forms an integrated "evolution-learning" loop: offspring morphologies are generated through mutation → parent control is inherited → a small number of PPO updates are applied → fitness is evaluated → selection and further mutation occur. Experiments on EvoGym multitasks demonstrate advantages over baselines like MLP-PPO and no inheritance in peak rewards, convergence speed, and robustness to morphological perturbations.

**Strengths:**

1. Using graphs to represent forms (nodes/edges) is a priori abstraction that makes the controller more robust to "node count/topology changes" and avoids the vulnerability of fixed input dimensions in MLP.

2. In response to the pain point of "training from zero as soon as the form changes", a mapweights inheritance (shared layer reuse, output layer channel alignment, new random/deletion dropout) is proposed, significantly reducing the cost of offspring warm-up.

**Weaknesses:**

1. Morphology generation relies on mutation and selection, lacking systematic integration of generative priors (VAE/diffusion/diffusion, microphysical guidance), and search efficiency may be limited in large morphological spaces. No control design with single-stage evolution has been achieved.

2. Comparison spectra are relatively concentrated: a systematic comparison between the main and other strong baselines of the graph structure (such as ModularEvoGym) and meta learning/fast adaptation methods.

**Questions:**

1. Can GAT representation and inheritance still generalize stably under irregular/hierarchical topology (joint chain, tree, graph mixture) and task migration settings? Are there any failed cases and diagnoses of 'negative transfer immediately after inheritance'?

2. Has the system been tested for stability under real-world factors such as large-scale morphological disturbances (large proportion of added or deleted nodes/switched neighborhoods), perceptual noise/delay, and execution saturation?

---

### Official Review · Reviewer_rGkR · 2025-11-02

**Soundness:** 2
**Presentation:** 2
**Contribution:** 1
**Rating:** 0
**Confidence:** 4

**Summary:**

This paper aims to implement a graph neural network–based co-design framework for soft robot morphology and control. Robots are represented as graphs, and a graph attention network combined with an MLP enables policies to update as the morphology changes.

**Strengths:**

+ This work aims to address a critical and well-recognized problem of morphology–control co-design in soft robots.
+  Scalability is an important challenge in soft robot co-design.

**Weaknesses:**

-  The novelty of this paper is rather limited. The primary contribution seems to be adding a graph network on top of an MLP to address scalability.
-  The problem formulation for morphology–control co-design is unclear. What exactly is the control space (e.g., motion or torque variables) that the method optimizes for control?
-  It is not clear why the proposed method uses a “decentralized structure” (Line 190), nor why it is inherently scalable to adding or removing actuators. For instance, if actuators are added, the control policy still needs to change to control the added motors (and the added DoF). Thus, the claim that “morphological changes such as adding or removing actuators can be incorporated without redesigning the policy” is not convincing.
-  How are morphologies generated or sampled? The paper provides no detail on the process of morphology design.
-  It is questionable to present this method as a co-design approach. The pipeline appears sequential: first generating a morphology, then optimizing a controller for that morphology. Soft robot co-design should tightly integrate morphology design with motion control and physical constraints; the current framework does not clearly demonstrate such coupling.
- The literature review is very limited. For example, key recent work in morphology-control co-design is largely omitted.
-  Experiments are limited to a simple 2-D discrete simulation. There is no evaluation on more realistic soft robot simulations or physical platforms, making generalizability to real soft robots questionable.
-  Experiments do not include comparisons with state-of-the-art methods, making it difficult to assess performance improvements or contributions relative to existing work.

**Questions:**

Please see the Weaknesses section.

---

### Meta-Review · Area_Chair_VufJ · 2026-01-06

**Summary:**

The reviewers have strong concerns about its contribution to the field. There is not much new materials in the method and the empirical results are limited.

**Reviewer Concerns:**

No rebuttal.

**Reviewer Scores:**

0000

---

### Decision · Program_Chairs · 2026-01-26

Reject